# Generalizing Off-Policy Learning under Sample Selection Bias

**Tobias Hatt**[1]      **Daniel Tschernutter**[1]      **Stefan Feuerriegel**[1,2]

[1]ETH Zurich, Switzerland
[2]LMU Munich, Germany

## Abstract

Learning personalized decision policies that generalize to the target population is of great relevance. Since training data is often not representative of the target population, standard policy learning methods may yield policies that do not generalize target population. To address this challenge, we propose a novel framework for learning policies that generalize to the target population. For this, we characterize the difference between the training data and the target population as a sample selection bias using a selection variable. Over an uncertainty set around this selection variable, we optimize the minimax value of a policy to achieve the best worst-case policy value on the target population. In order to solve the minimax problem, we derive an efficient algorithm based on a convex-concave procedure and prove convergence for parametrized spaces of policies such as logistic policies. We prove that, if the uncertainty set is well-specified, our policies generalize to the target population as they can not do worse than on the training data. Using simulated data and a clinical trial, we demonstrate that, compared to standard policy learning methods, our framework improves the generalizability of policies substantially.

## 1 INTRODUCTION

Learning personalized policies has become integral to modern decision-making in a variety of domains such as medicine [Hill and Su, 2013], and public policy [Kube et al., 2019]. Since in these domains exploration is costly or otherwise infeasible, many methods have been proposed for off-policy learning, i. e., policy learning from existing data [e. g., Dudík et al., 2014, Kallus, 2018, Athey and Wager, 2021, Tschernutter et al., 2022].

A major challenge in off-policy learning is the generalizability of policies. Generalizability is concerned with whether a policy learned on the data for training (i. e., training data) is also effective in the target population. Standard methods for policy learning yield policies that are effective on the target population, if, and only if, the training data is representative of the target population [e. g., Beygelzimer and Langford, 2009, Dudík et al., 2014]. However, this may not hold true in practice [e. g., Buchanan et al., 2018, Cole and Stuart, 2010, Downs and Black, 1998, Flores et al., 2021, Norris et al., 2001, Rothwell, 2005]. For instance, a review of HIV/AIDS clinical trials found that women are largely underrepresented in these trials [Gandhi et al., 2005, Greenblatt, 2011], so that data from these trials is not representative of the actual target population (i.e., the population of HIV-positive patients in the USA). Therefore, when data from such trials is used to derive policies, standard methods for policy learning may not generalize to the target population. As such, these policies may be ineffective or even harmful on the target population and, therefore, not relevant in practice.

In this paper, we develop a framework for learning policies from training data that generalize to the target population.[1] For this, we characterize the difference between training data and target population as a sample selection bias using an unknown selection variable [e. g., Cortes et al., 2008, Manski, 1989]. If we had oracle access to the true selection variable, we could re-weight the data accordingly in order to obtain the value of a policy on the target population. Since, in practice, the true selection variable is unknown, the value of a policy on the target population is not identifiable from training data. Instead, we derive bounds on the odds-ratio of the selection probability, which yields an uncertainty set around the true selection probabilities. Then, our framework optimizes the minimax value of a policy to achieve the best worst-case policy value on the target population. We prove that, if the uncertainty set is well-specified, our framework yields policies that do not do worse on the target population

---

[1]Code available at [github.com/tobhatt/GeneralOPL](github.com/tobhatt/GeneralOPL).

*Accepted for the 38th Conference on Uncertainty in Artificial Intelligence* (UAI 2022).

than the worst-case policy value estimated from the training data. As such, these policies can generalize to the target population. In order to efficiently optimize the minimax value of a policy, we show that it can be written as a difference of convex functions (DC) program. Then, by leveraging the structure of the adversarial subproblem, we develop a tailored **m**ini**m**ax **c**onvex-**c**oncave **p**rocedure (MMCCP). We prove that MMCCP converges for certain parameterized spaces of policies such as logistic policies. Using synthetic data and a clinical trial, we demonstrate that standard policy learning methods generalize poorly, while our framework improves the generalizability of policies substantially. As such, our framework enables to learn reliable policies that can be implemented in the target population.

## 2 PRELIMINARIES

In this section, we describe the setup, formulate the problem of generalizing policies, and discuss related work.

### 2.1 SETUP

We consider a binary treatment $T \in \{0, 1\}$, covariates $X \in \mathcal{X} \subseteq \mathbb{R}^d$, and the outcome $Y \in \mathbb{R}$. We use the convention that lower outcomes are preferred. Using the Neyman-Rubin potential outcomes framework [Rubin, 2005], let $Y(0), Y(1)$ be the potential outcomes for each of the treatments. Further, let a *policy* $\pi$ be a map from the covariates to the probability of treatment assignment, i.e., $\pi : \mathcal{X} \to [0, 1]$. Then, the *policy value* of $\pi$ is given by

$$V(\pi) = \mathbb{E}[Y^\pi] = \mathbb{E}[\pi(X)Y(1) + (1 - \pi(X))Y(0)]. \tag{1}$$

The objective of *policy learning* is to find a policy $\overline{\pi}$ in a policy class $\Pi$ that minimizes the policy value, i.e., $\overline{\pi} \in \arg\min_{\pi \in \Pi} V(\pi)$.

We make the following three standard assumptions [Rubin, 1974]: (i) consistency (i.e., $Y = Y(T)$); (ii) positivity (i.e., $0 < p(T = 1 \mid X = x) < 1$ for all $x$); and (iii) strong ignorability (i.e., $Y(0), Y(1) \perp\!\!\!\perp T \mid X$). Then we can identify the policy value in (1) in terms of the observed data $(X, T, Y)$.

### 2.2 PROBLEM FORMULATION

Suppose we are interested in learning a policy that minimizes the policy value under the target distribution $(X, T, Y) \sim \mathbb{P}$, i.e., $V_{\text{Target}}(\pi)$. However, we are *not* given data from the target distribution, but only data from a (potentially different) training distribution $(X, T, Y) \sim \mathbb{P}_{\text{Train}}$.

Standard policy learning methods assume that the training and target distributions are identical. However, even in carefully designed clinical trials, the subjects in the trial are often

not representative of the target population, i.e., $\mathbb{P}_{\text{Train}} \neq \mathbb{P}$ [e.g., Buchanan et al., 2018, Cortes et al., 2008, Downs and Black, 1998, Flores et al., 2021, Gandhi et al., 2005, Greenblatt, 2011, Rothwell, 2005]. Hence, standard methods for policy learning yield policies that minimize the policy value on the training data, i.e., $V_{\text{Train}}(\pi) = \mathbb{E}_{\text{Train}}[Y^\pi]$. However, since the policy value depends on the underlying data distribution, these policies may *not* minimize the policy value on the target population, i.e., $V_{\text{Target}}(\pi) = \mathbb{E}[Y^\pi]$. This can be seen when writing $\mathbb{E}[Y^\pi]$ in terms of the distribution $\mathbb{P}_{\text{Train}}$ using a change of probability measure:

$$\mathbb{E}[Y^\pi] = \mathbb{E}_{\text{Train}}[R Y^\pi], \tag{2}$$

where the random variable $R = \mathrm{d}\mathbb{P}/\mathrm{d}\mathbb{P}_{\text{Train}}$ is the *Radon-Nikodým derivative*,[2] also know as *density ratio*. As a direct implication, if $\mathbb{P}_{\text{Train}} \neq \mathbb{P}$ and, thus, $R \neq 1$, it follows that

$$\mathbb{E}[Y^\pi] \neq \mathbb{E}_{\text{Train}}[Y^\pi]. \tag{3}$$

In other words, a policy learned from training data using standard methods may not generalize to the target population, and, as such, may be of little help in practice.

In this paper, we consider the realistic setting in which the training data is *not* representative of the target population. We propose a framework for learning policies that generalize to the target population only given data from the training distribution, i.e., $\{(X_i, T_i, Y_i)\}_{i=1}^n \sim \mathbb{P}_{\text{Train}}$.

### 2.3 RELATED WORK

Despite the vast literature on off-policy learning, less work considers the problem of learning policies that generalize to the target population. Below, we summarize works on off-policy learning and works on external validity in causality, which is closely related to generalizability.

**Off-policy learning.** Off-policy learning methods can be broadly divided into three categories: (i) Direct methods estimate the outcome functions $\mu_t(x) = \mathbb{E}[Y(t) \mid X = x]$ and plug them into (1), i.e., $\hat{V}^{\text{DM}}(\pi) = \frac{1}{n} \sum_{i=1}^n \pi(X_i)\hat{\mu}_1(X_i) + (1 - \pi(X_i))\hat{\mu}_0(X_i)$ [e.g., Bennett and Kallus, 2020]. This approach is closely related to estimating the conditional average treatment effect, i.e., $\mathbb{E}[Y(1) - Y(0) \mid X]$ [Shalit et al., 2017, Hatt et al., 2022]. Direct methods are known to be weak against model misspecification with regards to $\mu_t(x)$. (ii) Weighting methods re-weight the outcome data such that it looks as if it were generated by the policy that is evaluated [e.g., Bottou et al., 2013, Horvitz and Thompson, 1952, Kallus, 2018, Li et al., 2011]. A common choice for weights are the normalized inverse propensity weights [Swaminathan and Joachims, 2015], i.e., $\hat{V}^{\text{NIPW}}(\pi) = \frac{1}{n} \sum_{i=1}^n 2W_i^{\text{IPW}}(1 -$

---

[2]The standard assumption that $\mathbb{P}$ is absolute continuous with respect to $\mathbb{P}_{\text{Train}}$, i.e., $\mathbb{P} \ll \mathbb{P}_{\text{Train}}$, is made in order to ensure that the Radon-Nikodým derivative is well-defined.

$2T_i)(1 - T_i - \pi(X_i))Y_i/(\frac{1}{n}\sum_{j=1}^{n} W_j^{\text{IPW}})$, where $W_i^{\text{IPW}} = 1/((1 - 2T_i)(1 - T_i - \pi^b(X_i)))$ and $\pi^b(x) = \mathbb{P}(T = 1 \mid X = x)$ is the so-called behavior policy, which was used to generate the training data. (iii) Doubly robust methods combine direct and weighting methods typically using the augmented inverse propensity weight estimator [Athey and Wager, 2021, Dudík et al., 2014, Thomas and Brunskill, 2016]. When the direct estimate of $\hat{\mu}_t$ is biased, the doubly robust method weights the residuals by the inverse propensity weights in order to remove the bias, i.e., $\hat{V}^{\text{DR}}(\pi) = \hat{V}^{\text{DM}}(\pi) + \frac{1}{n}\sum_{i=1}^{n} W_i^{\text{IPW}}(1 - 2T_i)(1 - T_i - \pi(X_i))(Y_i - \hat{\mu}_{T_i}(X_i))$.

The above methods have become the standard for off-policy learning. Despite their widespread use, the above methods implicitly assume that the training data, which is used to learn the policy, is representative of the target population. As such, when the training data is *not* representative of the target population, we cannot rely on the above methods, as policies may not generalize to the target population.

**Distributionally robust optimization.** A related, yet fundamentally different idea is distributionally robust optimization (DRO) [e.g., Duchi and Namkoong, 2018], which studies robustness towards distributional shifts. DRO has found application in off-policy learning by optimizing worst-case policy values [Si et al., 2020] and individualized treatment rules [Zhao et al., 2019b, Mo et al., 2020]. While generalizability is related to DRO, since the difference between training and target distribution can be seen as a distributional shift, it is fundamentally different, as DRO allows for arbitrary changes in distribution. In contrast, generalizability considers a training distribution that is, potentially, not representative of the target population, but derived from the target population. That is, generalizability considers differences in the distributions the arise from an unknown selection mechanism into the training data. Moreover, DRO and its applications require the decision-maker to quantify the distance between training and target distribution in terms of some divergence measure (typically the Kullback-Leibler divergence), which may be notoriously difficult for domain experts such as clinicians. In contrast, our approach allows for user-friendly and intuitive calibration of the parameters involved in the uncertainty set due to recognizing that the differences arises from an unknown selection mechanism.

**External validity in causality.** Different to policy learning, causal inference aims to estimate causal effects from observational data [Bottou et al., 2013, Kuzmanovic et al., 2021, Hatt and Feuerriegel, 2021]. External validity in causal inference is concerned with whether causal effect estimates obtained from a study sample are also valid for the target population. A common approach to address this is to re-weight the data with the inverse of a subject's probability to be selected into the study sample [e.g., Buchanan et al., 2018, Cole and Stuart, 2010, Dahabreh et al., 2019, Imai et al., 2013, Stuart et al., 2011]. This idea has been extended

to a doubly robust method for off-policy learning [Uehara et al., 2020]. Predominantly used in economics, the Heckman correction is another technique that is also based on a subject's selection probability [Heckman, 1979]. However, in order to estimate these selection probabilities, all existing approaches assume that data from both the study sample *and* the target population is given. In practice, however, we are only given data from the study sample and not from the target population. Other approaches include approximations of the bias arising from the difference in the study sample and target population by using weights that do not depend on the selection variable [Andrews and Oster, 2017], by bounding the weights directly [Aronow and Lee, 2013], or, in addition, by constraining the shape of the population outcome distribution [Miratrix et al., 2018].

Different to the above approaches and more practically, we do not assume that we have access to samples from the target population and, therewith, we cannot estimate a subject's selection probability. As a remedy, we present our framework for learning generalizable policies in the following.

## 3 GENERALIZING OFF-POLICY LEARNING UNDER SAMPLE SELECTION BIAS

In this section, we introduce our framework for learning policies that generalize to the target population. For this, we first characterize the difference between the training distribution $\mathbb{P}_{\text{Train}}$ and the target distribution $\mathbb{P}$ as a sample selection bias (Section 3.1). Then, based on this, we derive an uncertainty set and optimize the minimax policy value to achieve the best worst-case policy value (Section 3.2). We prove that policies learned in this way do not do worse on the target population than the worst-case policy value and, as such, can be generalized to the target population (Section 3.3).

### 3.1 SAMPLE SELECTION BIAS

In this section, we characterize the difference between the training distribution $\mathbb{P}_{\text{Train}}$ and the target distribution $\mathbb{P}$ as a sample selection bias using a selection variable [e.g., Cortes et al., 2008, Manski, 1989]. This then allows us to characterize the Radon-Nikodým derivative $R = \mathrm{d}\mathbb{P}/\mathrm{d}\mathbb{P}_{\text{Train}}$ in (2) in terms of the selection variable.

We represent the selection bias with a selection variable $S \in \{0, 1\}$. If, for a subject, $S = 1$, the subject is included in the training data, and, if $S = 0$, the subject is excluded from the training data. As a result, we can write the training distribution in terms of the target distribution:

$$\mathbb{P}_{\text{Train}}(\cdot) = \mathbb{P}(\cdot \mid S = 1). \tag{4}$$

Based on this, we characterize the Radon-Nikodým deriva-

tive, which enables us to write the policy value on the target population in terms of the selection variable $S$ and the training distribution $\mathbb{P}_{\text{Train}}$.

**Proposition 1.** *Under the sample selection bias, we can write the Radon-Nikodým derivative $R = d\mathbb{P}/d\mathbb{P}_{\text{Train}}$ as*

$$R = \frac{\mathbb{P}(S=1)}{\mathbb{P}(S=1 \mid X, T, Y)}, \qquad (5)$$

*and, therefore, we can write the policy value on the target population as*

$$V_{\text{Target}}(\pi) = \mathbb{E}_{\text{Train}}\left[\frac{\mathbb{P}(S=1)}{\mathbb{P}(S=1 \mid X, T, Y)} Y^{\pi}\right]. \qquad (6)$$

See Appendix A.1 for a proof. If, hypothetically, we observed $S$, we could estimate $R = \mathbb{P}(S=1)/\mathbb{P}(S=1 \mid X, T, Y)$[3] and re-weight the data accordingly in order to obtain the policy value on the target population. However, we never observe the selection variable $S$, since we only observe the training data for which $S = 1$. This renders the selection variable $S$ unidentifiable from the training data. Instead, we use an uncertainty set over which we optimize the minimax policy value on the target population.

## 3.2 LEARNING GENERALIZABLE POLICIES BY OPTIMIZING MINIMAX POLICY VALUE

We derive an uncertainty set around $R = \mathbb{P}(S=1)/\mathbb{P}(S=1 \mid X, T, Y)$ over which we maximize the policy value to obtain the worst-case policy value. Then, our framework optimizes the minimax policy value to achieve the best worst-case policy value on the target population.

If we had oracle access to the true Radon-Nikodým derivative $R_i^* = R_i^*(X_i, T_i, Y_i)$, we could estimate the policy value on the target population using Proposition 1, that is, by re-weighting the data with $R^*$. This often leads to high variance estimates due to probabilities close to zero. As a remedy, since $\mathbb{E}[R^*] = 1$, we use the empirical sum of the true Radon-Nikodým derivatives as a control variate to normalize the estimate. This gives rise to the following Hajek estimator for the policy value on the target population $V_{\text{Target}}(\pi)$:

$$\hat{V}_{\text{Target}}^*(\pi) = \frac{\sum_{i=1}^n R_i^* \psi_i(\pi)}{\sum_{i=1}^n R_i^*}, \qquad (7)$$

where $\psi_i(\pi)$ corresponds to one of the three standard methods for policy learning: direct, weighting, and doubly robust methods. Formally, $\psi_i(\pi)$ is either $\psi_i^{\text{DM}}(\pi)$, $\psi_i^{\text{NIPW}}(\pi)$, or

$\psi_i^{\text{DR}}(\pi)$ given as:

$$\psi_i^{\text{DM}}(\pi) = \pi(X_i)\mu_1(X_i) + (1 - \pi(X_i))\mu_0(X_i), \qquad (8)$$

$$\psi_i^{\text{NIPW}}(\pi) = \frac{2W_i^{\text{IPW}}(1 - 2T_i)}{\frac{1}{n}\sum_{j=1}^n W_j^{\text{IPW}}}(1 - T_i - \pi(X_i))Y_i, \qquad (9)$$

$$\psi_i^{\text{DR}}(\pi) = \psi_i^{\text{DM}}(\pi) + W_i^{\text{IPW}}(1 - 2T_i)(1 - T_i - \pi(X_i))(Y_i - \mu_{T_i}(X_i)). \qquad (10)$$

The outcome functions $\mu_t(x)$ and the weights $W^{\text{IPW}}$ need to be estimated from data. Any $\psi(\pi)$ in (8), (9), or (10) can be chosen for estimating the policy value as long as the data is re-weighted with the Radon-Nikodým derivative $R^*$.

Since the true $R^*$ is unknown, we instead derive a worst-case policy value on the target population. This allows to ensure that our policy does not do worse than expected once it is implemented in the target population. For this, we maximize (7) over an uncertainty set around $R^*$. We consider an uncertainty set motivated by sensitivity analysis in causality [e.g., Kallus et al., 2019, Kallus and Zhou, 2018, Rosenbaum, 2002, Zhao et al., 2019a], which restricts by how much $\mathbb{P}(S=1 \mid X, T, Y)$ can vary from $\mathbb{P}(S=1)$ via the odds-ratio characterization as follows:

$$\frac{1}{\Gamma} \leqslant \frac{\mathbb{P}(S=1)(1 - \mathbb{P}(S=1 \mid X, T, Y))}{\mathbb{P}(S=1 \mid X, T, Y)(1 - \mathbb{P}(S=1))} \leqslant \Gamma, \qquad (11)$$

where $\Gamma \geqslant 1$. For $\Gamma = 1$, we have equal probability of selection, i.e., $\mathbb{P}(S=1 \mid X, T, Y) = \mathbb{P}(S=1)$ and, thus, no difference between the training data and the target population. Larger values of $\Gamma$ allow for larger variation in the probabilities of selection. The bounded odds-ratio in (11) immediately yields an uncertainty set for the Radon-Nikodým derivative:

$$\mathcal{R} = \{R \in \mathbb{R}_+^n : l \leqslant R_i \leqslant u, \, \forall i\}, \qquad (12)$$

$$\text{where } l = \frac{1 - \mathbb{P}(S=1) + \Gamma\mathbb{P}(S=1)}{\Gamma}, \qquad (13)$$

$$u = \Gamma(1 - \mathbb{P}(S=1)) + \mathbb{P}(S=1). \qquad (14)$$

The uncertainty set $\mathcal{R}$ includes all Radon-Nikodým derivatives $R$ that satisfy the odds-ratio restriction in (11). For a given policy, we seek the maximum policy value on the target population among all Radon-Nikodým derivatives in the uncertainty set. This yields the following worst-case policy.

**Definition 1.** *(Worst-case policy value.) The worst-case policy value on the target population under the bounded odds-ratio with parameter $\Gamma$ is given by*

$$\overline{V}_{\text{Target}}(\pi; \mathcal{R}) = \max_{R \in \mathcal{R}} \frac{\sum_{i=1}^n R_i \psi_i(\pi)}{\sum_{i=1}^n R_i}, \qquad (15)$$

*where $\psi_i(\pi)$ corresponds to either (8), (9), or (10).*

---

[3]Under the standard assumption $\mathbb{P} \ll \mathbb{P}_{\text{Train}}$, the selection variable satisfies positivity, i.e., $\mathbb{P}(S=1 \mid X, T, Y) > 0$. Therefore, the ratio $\mathbb{P}(S=1)/\mathbb{P}(S=1 \mid X, T, Y)$ is well-defined.

Then, we seek the optimal policy in a policy class $\Pi$, which minimizes the worst-case policy value on the target population, i. e.,

$$\overline{\pi}(\Pi, \mathcal{R}) \in \underset{\pi \in \Pi}{\arg\min}\ \overline{V}_{\text{Target}}(\pi; \mathcal{R}). \qquad (16)$$

In particular, a policy learned with our framework generalizes to the target population, since it does not do worse on the target population than the worst-case policy value estimated using the training data. For this, a decision-maker only has to quantify the population selection probability, i. e., $\mathbb{P}(S = 1)$ and appropriately choose the maximum deviation from it via $\Gamma$. We discuss data-driven approaches to choose these quantities in Section 3.4. We derive a tailored convex-concave procedure for optimizing (16) in Section 4.

### 3.3 THEORETICAL GUARANTEES FOR GENERALIZABILITY

We prove that, if the Radon-Nikodým is appropriately bounded, the worst-case policy value, $\overline{V}_{\text{Target}}(\pi; \mathcal{R})$, is asymptotically an upper bound for the true policy value on the target population, $V_{\text{Target}}(\pi)$. As such, a policy learned with our framework does not do worse on the target population than the worst-case policy value. Similar to [Athey and Wager, 2021], we express the flexibility of a policy class $\Pi$ using the notion of the Rademacher complexity, i. e., $\mathcal{R}_n(\Pi)$.[4]

**Theorem 1.** *(Generalization bound.) Suppose the true Radon-Nikodým derivative is appropriately bounded, i. e., $R^* \in \mathcal{R}$ and, therefore, $l \leqslant R_i^* \leqslant u$, and we have bounded outcomes, i. e., $|Y| < C$. Then, for a constant $K^\psi$ depending on $\psi(\pi)$ and for some $\delta > 0$, we have that,*

$$V_{Target}(\pi) \leqslant \overline{V}_{Target}(\pi; \mathcal{R}) + 2C \frac{u}{l} K_\psi \left( \mathcal{R}_n(\Pi) + \sqrt{\frac{18 \log(4/\delta)}{n}} \right),$$
$$(17)$$

*with probability at least $1 - \delta$ and for any $\pi \in \Pi$.* $\qquad \square$

See Appendix A.2 for a proof. All policy classes we consider have $\sqrt{n}$-vanishing Rademacher complexity, i. e., $\mathcal{R}_n(\Pi) = \mathcal{O}(n^{-1/2})$. Therefore, Theorem 1 proves that, asymptotically, $\overline{V}_{\text{Target}}(\pi)$ is an upper bound for $V_{\text{Target}}(\pi)$. This guarantees that $\overline{\pi}(\Pi, \mathcal{R})$ from (16) does not do worse on the target population than the worst-case policy value, which is calculated using training data. In particular, since $\overline{\pi}(\Pi, \mathcal{R})$ minimizes the right hand side of (17), $\overline{\pi}(\Pi, \mathcal{R})$ is the best policy that guarantees to generalize to the target population. Our bound in Theorem 1 holds without complete knowledge of the selection variable and proves that our framework yields policies that generalizes to the target population.

Note that in Theorem 1, we use the true nuisance functions instead of estimates, since it has been shown that this does not affect the leading term in the convergence rate of the policy value (see Athey and Wager [2021]; Sec. 3.1, Sec. 3.2, and Lemma 4). This holds true if the nuisance functions have finite second moment and we use consistent estimators for the nuisance functions and $L^2$ errors decay with $1/n^\zeta$, where $\zeta$ depends on the nuisance functions. Hence, to provide a generalization bound on the policy value, it is enough to consider the true nuisance functions as we did in Theorem 1.

### 3.4 CALIBRATION OF $\Gamma$ AND $\mathbb{P}(S = 1)$

In this section, we discuss two approaches to calibrate the parameters $\Gamma$ and $\mathbb{P}(S = 1)$ in (11), which are context-dependent: (i) Practitioner calibration with domain knowledge and (ii) data-driven calibration.

**(i) Practitioner calibration:** This approach is based on domain knowledge of practitioners about variables that impact selection into training data. First, $\mathbb{P}(S = 1)$, the population probability of inclusion, needs to be quantified. If the study is randomized, a value $\approx 1/2$ is reasonable. Second, $\Gamma$, the largest deviation from $\mathbb{P}(S = 1)$, needs to be quantified. Our framework allows a practitioner-friendly choice of calibration parameters. In fact, both questions may be simply answered using domain knowledge.

**(ii) Data-driven calibration:** Although our framework enables practitioners to choose appropriate calibration parameters, we provide a fully data-driven approach for calibrating $\Gamma$ and $\mathbb{P}(S = 1)$. To this end, we consider a setting in which samples from *one* of the covariates of the target population are provided. This is reasonable, since we often have limited understanding of the target population and, for instance, know covariates such as the distribution of gender or age in the target population. Once we are given one covariate, e. g., $x_{\text{age}}$, we proceed in two steps: (1) For calibrating $\mathbb{P}(S = 1)$, we approximate $\mathbb{P}(S = 1 \mid X, Y, T)$ via an estimate of $\mathbb{P}(S = 1 \mid x_{\text{age}})$ and, based on this, we approximate $\mathbb{P}(S = 1)$ by averaging over $x_{\text{age}}$, i. e., $\frac{1}{n}\sum_{i=1}^{n} \mathbb{P}(S_i = 1 \mid x_{\text{age},i})$. (2) For calibrating $\Gamma$, we take the maximum of the odds-ratio in (11) with the above estimates for $\mathbb{P}(S = 1)$ and $\mathbb{P}(S = 1 \mid x_{\text{age}})$ plugged in, which yields a value for $\Gamma$. We use this data-driven calibration procedure in our experiments (Section 5).

In case the uncertainty regarding the calibration parameters remains high, large values for $\Gamma$ can be chosen, yielding a wide uncertainty set and conservative policies.

## 4 OPTIMIZING GENERALIZABLE POLICIES

In this section, we derive an efficient algorithm for optimizing the minimax policy value in (16). For this, we consider

---

[4]The empirical Rademacher complexity of a policy class $\Pi$ is defined as $\mathcal{R}_n(\Pi) = \frac{1}{2^n}\Sigma_{\sigma\in\{-1,+1\}^n}\sup_{\pi\in\Pi}|\frac{1}{n}\sum_{i=1}^{n}\sigma_i\pi(X_i)|$.

a parameterized policy class $\Pi = \{\pi(\cdot, \theta) : \theta \in \Theta\}$ and the minimax problem

$$\min_{\theta \in \Theta} \max_{R \in \mathcal{R}} \frac{\sum_{i=1}^{n} R_i \psi_i(\theta)}{\sum_{i=1}^{n} R_i}, \qquad \text{(MMP)}$$

where $\psi_i(\theta)$ denotes $\psi_i(\pi(\cdot, \theta))$ and corresponds to either (8), (9), or (10). The above minimax problem is non-trivial, since it is in general non-convex in $\theta$. We first derive a closed-form solution of the worst-case policy value subproblem (Section 4.1). Then, based on this, we develop a tailored convex-concave procedure that solves (MMP) (Section 4.2).

## 4.1 CLOSED-FORM SOLUTION OF WORST-CASE POLICY VALUE

The solution of (MMP) involves the worst-case policy value subproblem in (15). We derive a closed-form solution of the subproblem and the corresponding Radon-Nikodým derivative at the optimal solution in Theorem 2.

**Theorem 2.** *(Closed-form solution of worst-case policy value.) Let $(i)$ denote the ordering such that $\psi_{(1)}(\theta) \leqslant \ldots \leqslant \psi_{(n)}(\theta)$. Then, an optimal solution of the worst-case policy value subproblem (15) is given by*

$$\overline{V}_{Target}(\pi; \mathcal{R}) = \frac{l \sum_{i=1}^{k^*} \psi_{(i)}(\theta) + u \sum_{i=k^*+1}^{n} \psi_{(i)}(\theta)}{lk^* + u(n - k^*)},$$
$$\text{(18)}$$

*with*

$$k^* = \inf \Big\{ k \in \{0, \ldots, n\} : \qquad\qquad\text{(19)}$$
$$\frac{l \sum_{i=1}^{k} \psi_{(i)}(\theta) + u \sum_{i=k+1}^{n} \psi_{(i)}(\theta)}{lk + u(n - k)} \leqslant \psi_{(k+1)}(\theta) \Big\}. \quad \text{(20)}$$

*The Radon-Nikodým derivative at optimal solution is given by $R_{(i)} = l\mathbb{1}\{(i) \leqslant k^*\} + u\mathbb{1}\{(i) > k^*\}$.*

See Appendix A.3 for a proof. Theorem 2 is appealing for two reasons: (i) We prove that $\overline{V}_{Target}(\pi; \mathcal{R})$ is efficiently solved by a linear search over the sorted data. (ii) We prove that the worst-case policy value is given by a maximum over a finite set, which we use in the following section to show that the minimax problem can be written as a difference-of-convex functions (DC) problem. Based on this, we develop a convex-concave procedure to efficiently solve the minimax problem in (MMP).

## 4.2 MINIMAX CONVEX-CONCAVE PROCEDURE

In this section, we develop the **m**ini**m**ax **c**onvex-**c**oncave **p**rocedure (MMCCP) to efficiently solve the minimax problem (MMP). For this, we derive a DC-representation of the worst-case policy value based on its closed-form solution in Theorem 2. For this, the following assumptions are made.

**Assumption 1.** *The set $\Theta$ is nonempty, compact, and convex. Furthermore, $\pi$ is a DC-function in $\theta$, i. e., $\pi(X, \theta) = \tilde{g}(X, \theta) - \tilde{h}(X, \theta)$, where $\tilde{g}$ and $\tilde{h}$ are convex in $\theta$, and differentiable.*

Note that Assumption 1 is very general as the class of DC-functions is very rich. For instance, it includes all twice continuously differentiable functions [Horst and Thoai, 1999]. We later show that Assumption 1 is fulfilled for the established policy class of logistic policies. First, we show that $\psi_i(\theta)$ can be written as a DC-function.

**Lemma 1.** *(DC-representation of $\psi_i(\theta)$.) Under Assumption 1, $\psi_i(\theta)$ is a DC-function in $\theta$, i. e.,*

$$\psi_i(\theta) = g_i(\theta) - h_i(\theta), \qquad\qquad (21)$$

*where $g_i$ and $h_i$ are convex in $\theta$.*

See Appendix A.4 for a proof. Now, using Lemma 1 and Theorem 2, we prove that the worst-case policy value can be written as a DC-function.

**Theorem 3.** *(DC-representation of worst-case policy value.) Under Assumption 1, the worst-case policy value $\overline{V}_{Target}(\pi; \mathcal{R})$ is a DC-function in $\theta$, i. e.,*

$$\overline{V}_{Target}(\pi; \mathcal{R}) = g(\theta) - h(\theta), \qquad (22)$$

*where $g(\theta)$ and $h(\theta)$ are convex and given by*

$$g(\theta) = \max_{R \in \mathcal{R}} \frac{\sum_{i=1}^{n} R_i \psi_i(\theta)}{\sum_{i=1}^{n} R_i} + \sum_{i=1}^{n} h_i(\theta) c_i, \qquad (23)$$

$$h(\theta) = \sum_{i=1}^{n} h_i(\theta) c_i, \qquad\qquad (24)$$

*with $g_i$ and $h_i$ from Lemma 1, and non-negative constants $c_i$ for all $i$.*

See Appendix A.5 for a proof. Finally, with the DC-representation of the worst-case policy value in Theorem 3, we can write the original minimax problem in (MMP) as a DC-program, i. e.,

$$\min_{\theta \in \Theta} g(\theta) - h(\theta), \qquad\qquad (25)$$

where $g(\theta)$ and $h(\theta)$ are convex and given in Theorem 3. Hence, we can solve the minimax problem via a convex-concave procedure [Sriperumbudur and Lanckriet, 2009, Yuille and Rangarajan, 2003]. This yields our tailored MMCCP for solving (MMP) as outlined in Algorithm 1. Next, we prove that the sequence $(\theta^k)_{k \in \mathbb{N}}$ generated by MMCCP yields monotonically decreasing worst-case policy values and converges under mild assumptions.

**Theorem 4.** *(Theoretical Analysis of MMCCP.) Suppose the outcomes are bounded, i. e., $|Y| < C$, and Assumption 1 holds. Then, the following holds true:*

---

**Algorithm 1** MMCCP

---

**Input:** Initial theta $\theta^0$, convergence tolerance $\delta_{\text{tol}}$
Set $k \leftarrow 0$
**repeat**
    Solve the *convex* problem:
$$\theta^{k+1} \quad \in \quad \arg\min_{\theta \in \Theta} \max_{R \in \mathcal{R}} \frac{\sum_{i=1}^n R_i \psi_i(\theta)}{\sum_{i=1}^n R_i} \quad +$$
$$\sum_{i=1}^n c_i (h_i(\theta) - \langle \theta, \nabla h_i(\theta^k) \rangle)$$
    Set $k \leftarrow k + 1$
**until** $\|\theta^k - \theta^{k-1}\| < \delta_{\text{tol}}$

---

1. *The sequence* $(\theta^k)_{k \in \mathbb{N}}$ *generated by MMCCP satisfies the monotonic descent property, i.e., for all* $k \in \mathbb{N}$,

$$\max_{R \in \mathcal{R}} \frac{\sum_{i=1}^n R_i \psi_i(\theta^{k+1})}{\sum_{i=1}^n R_i} \leqslant \max_{R \in \mathcal{R}} \frac{\sum_{i=1}^n R_i \psi_i(\theta^k)}{\sum_{i=1}^n R_i}. \tag{26}$$

2. *If* $\tilde{g}$ *and* $\tilde{h}$ *from Assumption 1 are strongly convex,[5] then every limit point* $\theta^*$ *of* $(\theta^k)_{k \in \mathbb{N}}$ *is a stationary point[6] of* (MMP). *Furthermore, it holds:* $\lim_{k \to \infty} \|\theta^{k+1} - \theta^k\| = 0$.

See Appendix A.6 for a proof. To summarize, we develop a tailored convex-concave procedure that efficiently solves (MMP). This is only possible since we proved that the worst-case policy value has a DC-representation (see Theorem 3). In particular, our algorithm can be used on a rich class of policies and converges under mild assumptions. We now demonstrate that Assumption 1 holds for an established parameterized policy class which we use in our experiments.

**Logistic policies:** Logistic policies are defined by $\pi(X, \theta) = \sigma(\theta^\intercal X)$, where $\sigma(z) = 1/(1 + e^{-z})$. To find a DC-representation, it is sufficient to decompose $\sigma(z)$. Hence, we set $z = \theta^\intercal X$ and write

$$\tilde{g}_{\log}(z) = \begin{cases} \frac{1}{4}z + \frac{1}{2}, & \text{if } z \geqslant 0, \\ \frac{1}{2}\tanh(\frac{1}{2}z) + \frac{1}{2}, & \text{else,} \end{cases} \tag{27}$$

$$\tilde{h}_{\log}(z) = \begin{cases} \frac{1}{4}z - \frac{1}{2}\tanh(\frac{1}{2}z), & \text{if } z \geqslant 0, \\ 0, & \text{else.} \end{cases} \tag{28}$$

It is straightforward to check that both functions are convex. They can be made strongly convex by adding $\frac{\lambda}{2}z^2$ to both functions. Since $\tilde{g}_{\text{Log}}$ and $\tilde{h}_{\text{Log}}$ are differentiable, Assumption 1 is fulfilled and, hence, MMCCP converges for logistic policies. In Appendix C, we show that Assumption 1 also holds for linear policies. In addition, logistic

---

[5]A function $f$ is strongly convex, if $\rho(f) > 0$, where $\rho(f)$ is the modulus of strong convexity of a convex function $f$, which is defined as $\rho(f) = \sup\{\rho \geqslant 0 : f(\cdot) - \frac{\rho}{2}\|\cdot\|_2^2 \text{ is convex}\}$.

[6]Note that the objective function is in general not differentiable, see Appendix D. Hence, we consider stationary points in the context of convex analysis, i.e., $0 \in \partial g(\theta^*) \cap \partial h(\theta^*)$, where $\partial$ denotes the subgradient.

policies also satisfy the generalization bound in Theorem 1, since they have $\sqrt{n}$-vanishing Rademacher complexity. This can be seen by using that $\sigma$ is Lipschitz together with the Rademacher bound for linear classes [Maurer, 2006] and the scalar concentration inequality [Maurer, 2016].

## 5 EXPERIMENTS

In this section, we compare several policy learning methods to policies learned with our framework on the example of logistic policies. We demonstrate that our framework generalizes substantially better to the target population.

### 5.1 SIMULATION STUDY

We first consider a simulation study to demonstrate the effect of unrepresentative training data. For this, we consider the following data-generating process for the target population:

$$\mathbf{X} \sim \mathcal{N}_5(\mu, \mathbf{I}_5), \quad T \mid \mathbf{X} \sim \text{Bern}(1/2), \tag{29}$$
$$Y \mid (\mathbf{X}, T) = m(\mathbf{X}) + T \cdot C(\mathbf{X}) + \epsilon, \tag{30}$$

where $m(\mathbf{X}) = \beta_0^\intercal \mathbf{X} + 3\xi$, $C(\mathbf{X}) = 5/2 + \beta_1^\intercal \mathbf{X} - 4\xi$, $\xi \sim \text{Bern}(1/2)$, and $\epsilon \sim \mathcal{N}(0, 1)$. The covariate mean is $\mu = [-1, 1/2, -1, 0, -1]$ and the outcome means are $\beta_0 = [0, 3/4, -1/2, 0, -1]$ and $\beta_1 = [-3/2, 1, -3/2, 1, 1/2]$, respectively. We sample from the target population using the following selection variable

$$S \sim \text{Bern}\left(\frac{1}{2} + \frac{0.95}{2}\tanh(-10\,C(\mathbf{X}))\right), \tag{31}$$

which yields training data that is unrepresentative for the tail of the covariate distribution.[7] As baselines, we consider three established policy learning methods: the direct method (**DM**), the normalized inverse propensity weights method (**NIPW**), and the doubly robust method (**DR**). We compare these established methods against our generalizable methods with each of the three $\psi(\theta)$ in (8), (9), and (10): the worst-case policy value obtained with the direct method (**GenDM**), obtained with the normalized IPW (**GenNIPW**), and with the doubly robust method (**GenDR**). We use kernel and logistic regression for estimating $\mu_t(x)$ and $W^{\text{IPW}}$. The parameter $\mathbb{P}(S = 1)$ is chosen by the data-driven calibration in Section 3.4 and $\Gamma$ is varied across $\{1.0, 1.2, 1.4, 1.6, 1.8, 2.0, 3.0, \ldots, 10.0\}$. Details on implementation of MMCCP are in Appendix D.

We present the results for the different values of $\Gamma$ in Figure 1. Specifically, we show by how much our methods improve over the policy regret, $V_{\text{Target}}(\hat{\pi}) - V_{\text{Target}}(\pi^*)$, of the corresponding baseline policy (i.e., DM, NIPW, and DR) when tested on the target population. Our methods achieve lower policy regrets on the target population across

---

[7]Note that we multiply the second term in (31) with $\frac{0.95}{2}$ to ensure that the selection probability remains strictly positive.

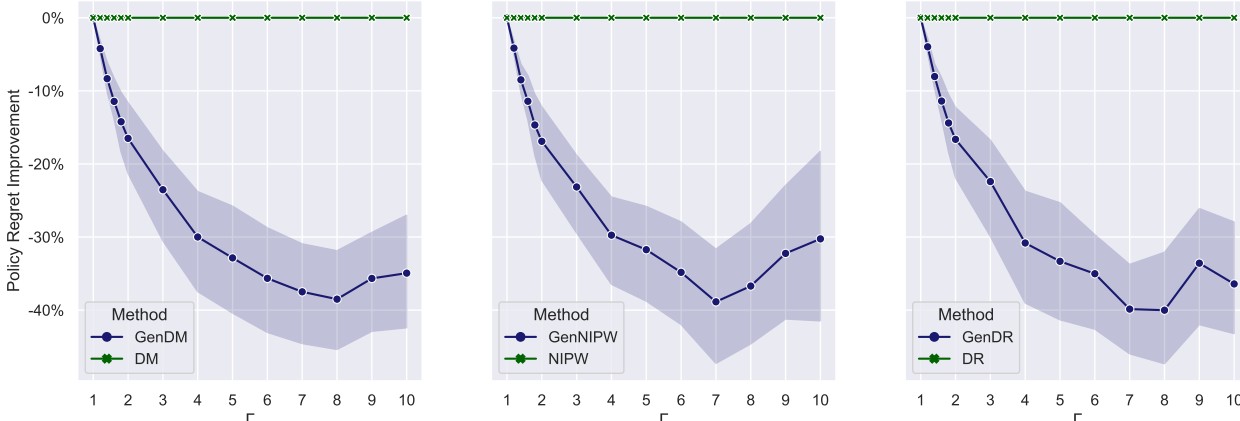

Figure 1: Policy regret improvement of our methods (blue) over the baseline methods (green) on the target population for different values of $\Gamma$. Compared to the baseline methods (i. e., DM, NIPW, and DR), our methods (i. e., GenDM, GenNIPW, and GenDR) show superior generalizability and, as such, improve the policy regret by up to $40\%$ at the true $\Gamma^* = 8$. Lower is better.

all methods and across all values of $\Gamma$. Specifically, relative to the policy regret of the baseline policy (green line), our methods (blue line) improve the policy regret on the target population by up to $40\%$. By construction, for $\Gamma = 1$ (left end of plots), our methods resemble the baseline methods and yield the same policy regret. When we increase $\Gamma$, our policies achieve substantial improvements of the policy regret on the target population over the baselines. The best policy regret on the target population is achieved for $\Gamma = 8$, which is consistent with the simulation specifications, as the true $\Gamma^* = 8$. For $\Gamma = 8$, relative to the baseline policies, our methods improve the policy regret by up to $40\%$. This demonstrates that policies learned with our framework generalize substantially better to the target population.

### 5.2 EXPERIMENTS ON CLINICAL TRIAL DATA

We evaluate our methods using the AIDS Clinical Trial Group (ACTG) study 175 [Hammer et al., 1996], which is particularly suited for evaluating our framework. This is because HIV-positive females tend to be underrepresented, which makes these studies not representative of the target population (i. e., the HIV-positive population in the USA) [Gandhi et al., 2005, Greenblatt, 2011]. In fact, in the ACTG 175 study, only $5.8\%$ of the patients are female, whereas HIV-positive females are more common in the USA population. The outcome $Y$ is the difference between the cluster of differentiation 4 (CD4) cell counts at the beginning of the study and the CD4 counts after $20 \pm 5$ weeks. The average treatment effects on the male and female subgroups are -8.97 and -1.39, respectively, suggesting a large discrepancy in treatment effects between both subgroups. We consider two treatment arms: one treatment arm for both zidovudine (ZDV) and zalcitabine (ZAL) ($T = 1$) vs. one treatment arm

for ZDV only ($T = 0$), comprising $1,056$ patients in total. We consider 12 covariates (details on the covariates are in Appendix B). Again, we compare our methods against the established baseline methods. This is a real-world clinical trial and, hence, we cannot access the true policy values on the target population. However, we investigate the behavior of our policies by studying the percentage of patients that are treated (i. e., $\pi(X) > 0.5$) for varying $\Gamma$. For our GenDR, the result is presented in Figure 2. The results for GenDM and GenNIPW are in Appendix E. We find that, compared to the baseline policy, our policy treats fewer patients for increasing $\Gamma$. This seems reasonable, since females are underrepresented and have a lower average treatment effect. Specifically, the baseline policy tends to treat more patients, since there are more patients in the study that benefit from the treatment. However, in the target population (with a greater proportion of females), fewer patients are expected to benefit (due to the lower treatment effect in the female subgroup). Our policy accounts for the underrepresentation of females and, as such, tends to treat fewer patients. This result indicates the potential of our framework for learning policies that generalize to the target population.

### 6 CONCLUSION

We propose a novel framework for learning policies that generalize to the target population by optimizes the minimax policy value on the target population. We prove that our framework yields policies that do not do worse on the target population than the worst-case policy value. We solve the minimax problem via a tailored convex-concave procedure for which we prove convergence for parametrized spaces of policies. Experiments demonstrate the benefit of learning generalizable policies using our framework.

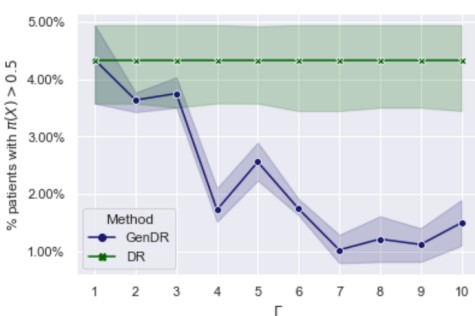

Figure 2: Percentage of patients with $\pi(X) > 0.5$ for our GenDR policy method. Fewer patients are treated for increasing $\Gamma$.

## ACKNOWLEDGEMENTS

We thank the anonymous reviewers for valuable feedback. This work was supported by the Swiss National Science Foundation grants number 186932.

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
