# OpenReview forum: "Generalizing Off-Policy Learning under Sample Selection Bias"
_auai.org/UAI/2022/Conference — UAI 2022 Poster_

### Official Review · Reviewer_LtFb · 2022-03-31

**Q2(1) Originality/Novelty:** 3
**Q2(2) Significance/Impact:** 2
**Q2(3) Correctness/Technical Quality:** 3
**Q2(6) Clarity Of Writing:** 3
**Q6 Overall Score:** 6
**Q8 Confidence In Your Score:** 3

**Q1 Summary And Contributions:**

This paper studies policy learning with covariate shift. Instead of estimating the degree of shift, it solves a minimax problem to obtain a robust policy, within a range controlled by a tuning parameter. The policy value is estimated via standard methods and the optimization among a parametric class is done by a proposed convex-concave algorithm.
Generalization bound is proven and heuristic tuning methods are discussed.
Numerical experiments show the improvement on value over base methods.

**Q10 Ethical Concerns (Optional):**

I do not have any major concerns.


**Q2 Assessment Of The Paper:**

More detailed information regarding each of these aspects is given below:

**Q2(4) Quality Of Experiments (Optional):**

3: Good: The experimental evaluation is adequate, and the results convincingly support the main claims.

**Q2(5) Reproducibility:**

3: Good: Key resources (e.g., proofs, code, data) are available and key details (e.g., proofs, experimental setup) are sufficiently well-described for competent researchers to confidently reproduce the main results.

**Q3 Main Strengths:**

1. The paper studies a novel problem in off-policy optimizaiton, with potential practical imporatnce.
2. The overall methodology design is sound.
3. The paper is well written and easy to understand.


**Q4 Main Weakness:**

My main suggestion is to mention and compare with other approaches of off-policy optimization in your introduction as well as experiments.
Specifically, the paper focuses on value-based policy search (among a pre-specified policy class), which will suffer the selection bias as the value is defined as an expectation over the covariate shift.
However, e.g., the so-call Q-learning in DTR (estimating E(Y|X,T) first and then take action greedily) will not have such an issue.


**Q5 Detailed Comments To The Authors:**

Except for the suggestion in Q4, I have several minor suggestions regarding the experiment section that I hope to see in the response.

1. Regarding the simulation setting, discussion on how significant the selection bias is and a study on how does it affect the improvement.
2. Add a skyline that knows the selection bias. It provides a more detailed picture of the trade-off from robustness.
3. Discussion and possibly more experiments on complexer policy classes (e.g., neural network). Is is possible to extend to those cases at all?


Besides, the argument on the top of page 5 (right column) is loose. A more rigorous argument or proof should be added (e.g., even in the appendix, mention where does this condition appear and what may happen when add the estimation errors there).


**Q7 Justification For Your Score:**

Overall, I think this paper is a nice contribution to the off-policy learning area.
It provides a technically sound solution to a practical problem, with theoretical and empirical supports.
My current evaluation is 6 as there are a few points which I mentioned above that one needs to see first to measure the significance.
I am willing to raise the score if they are addressed.


**Q9 Complying With Reviewing Instructions:**

1: Yes.

---

### Official Review · Reviewer_khWu · 2022-04-09

**Q2(1) Originality/Novelty:** 2
**Q2(2) Significance/Impact:** 2
**Q2(3) Correctness/Technical Quality:** 3
**Q2(6) Clarity Of Writing:** 3
**Q6 Overall Score:** 6
**Q8 Confidence In Your Score:** 3

**Q1 Summary And Contributions:**

To bridge the gap between the training data and target population, this paper proposed a novel framework for learning policies that generalize to the target population. By introducing a sample selection bias, the authors optimize the minmax value of the a policy to achieve the best worst-case policy on the target population. Furthermore, the authors derive an efficient algorithm based on a convex-concave procedure and prove convergence for certain parametrized space of policies.

**Q2 Assessment Of The Paper:**

More detailed information regarding each of these aspects is given below:

**Q2(4) Quality Of Experiments (Optional):**

2: Fair: The experimental evaluation is weak: important baselines are missing, or the results do not adequately support the main claims.

**Q2(5) Reproducibility:**

3: Good: Key resources (e.g., proofs, code, data) are available and key details (e.g., proofs, experimental setup) are sufficiently well-described for competent researchers to confidently reproduce the main results.

**Q3 Main Strengths:**

1. This paper proposed a novel framework to generalize off-policy learning under sample selection bias.
2. The authors derive min-max optimization formulation to learn generalizable policies.
3. This paper provides theoretical guarantees for the generalization bound of policy value.

**Q4 Main Weakness:**

The experiments may not be adequate to strongly support this paper's claims, especially for the experiment with real clinical data.
1. The experiment has a very limited number of samples, which makes the conclusion less convincing.
2. Can the authors add policy regret improvements compared to baselines to demonstrate their better policy generalization.

**Q5 Detailed Comments To The Authors:**

Can the author further reduce the variance for the target policy in Eq. 7? E.g. A. Owen and Y. Zhou, “Safe and effective importance sampling,”Journal of the American Statistical Association. In the experiment section, is that possible for the authors to investigate the variance of the target policy value estimation?

**Q7 Justification For Your Score:**

The authors propose a novel framework to generalize off-policy learning under sample selection bias, by optimizing the minimax policy value on the target population. This paper provides theoretical guarantees for the generalization bound of policy value. The experiments may not be adequate to strongly support this paper's claims, especially for the experiment with real clinical data.


**Q9 Complying With Reviewing Instructions:**

1: Yes.

---

### Official Review · Reviewer_Y1nz · 2022-04-12

**Q2(1) Originality/Novelty:** 3
**Q2(2) Significance/Impact:** 2
**Q2(3) Correctness/Technical Quality:** 3
**Q2(6) Clarity Of Writing:** 3
**Q6 Overall Score:** 6
**Q8 Confidence In Your Score:** 3

**Q1 Summary And Contributions:**

This paper learns the problem of learning policies that generalize to the target population. They characterize the difference between training data and the target population as a sample selection bias using an unobserved selection variable. They derive bounds on the odds-ratio of selection probability and proposed a framework that optimizes the minimax value of a policy to achieve the best worst-case policy value on the target population.

**Q2 Assessment Of The Paper:**

More detailed information regarding each of these aspects is given below:

**Q2(4) Quality Of Experiments (Optional):**

3: Good: The experimental evaluation is adequate, and the results convincingly support the main claims.

**Q2(5) Reproducibility:**

3: Good: Key resources (e.g., proofs, code, data) are available and key details (e.g., proofs, experimental setup) are sufficiently well-described for competent researchers to confidently reproduce the main results.

**Q3 Main Strengths:**

This paper studies a new problem of generalizing the policy to a target population, which is considered in the previous literature.
The paper develops a method to optimize the minimax value of a policy that achieves the best worst-case policy value on the target population with theoretical guarantee.
The experiments show that the proposed framework improves the generalizability of policies.


**Q4 Main Weakness:**

The proposed method needs to calibrate the parameters Γ and P(S=1). However, calibrating these two parameters needs some additional information or domain knowledge, which makes the proposed method not practical in the real-world applications.

In the experiments part, the author should show that the data-driven way of calibrating Γ is effective (near the true Γ*).

**Q5 Detailed Comments To The Authors:**

See Q3 and Q4

**Q7 Justification For Your Score:**

This paper provides a new idea for generalizing the policy to the target population. Although the proposed method has some limitations since it requires some extra information about the distribution of the target population, this paper still provides a valid and theoretically supported method to solve the challenging problem of selection bias with unknown selection variable.

**Q9 Complying With Reviewing Instructions:**

1: Yes.

---

### Official Review · Reviewer_vc8h · 2022-04-16

**Q2(1) Originality/Novelty:** 3
**Q2(2) Significance/Impact:** 3
**Q2(3) Correctness/Technical Quality:** 2
**Q2(6) Clarity Of Writing:** 3
**Q6 Overall Score:** 5
**Q8 Confidence In Your Score:** 3

**Q1 Summary And Contributions:**

This paper proposes to use a minimax learning framework to learn an optimal policy from off-policy data. The min player is the parameters of the policy while the max player is a weight, such that the objective is the normalized weighted sum of expected rewards on the training data. The motivation of this form of objective is to use the empirical sum Radon-Nikodým derivatives to normalize the expected rewards.  The method is tested on both synthetic data and real world data.


**Q2 Assessment Of The Paper:**

More detailed information regarding each of these aspects is given below:

**Q2(4) Quality Of Experiments (Optional):**

2: Fair: The experimental evaluation is weak: important baselines are missing, or the results do not adequately support the main claims.

**Q2(5) Reproducibility:**

4: Excellent: Key resources (e.g., proofs, code, data) are available and key details (e.g., proof sketches, experimental setup) are comprehensively described for competent researchers to confidently and easily reproduce the main results.

**Q3 Main Strengths:**

1. Using the minimax framework to optimize the empirical sum Radon-Nikodým derivatives on the expected rewards is somewhat novel.

2. The paper is generally well written and easy to understand.

3. This paper provides a general generalization bound and a specific learning algorithm when policy is parameterized.


**Q4 Main Weakness:**

1. The method regards DM, IPS, and DR as a black box, but actually there could be a lot of connections between the general framework and the standard methods. For example, IPS also has a weight, which is similar to the R variable. The framework completely ignores the details in these estimators and just wrap a minimax framework on top of them. I think this may not be the optimal way. The paper also does not discuss this.

2. The paper claims that the parameter in the constraint is easy to pick. But the experiment section only tries a large range of the parameter and shows the performance is best when the parameter matches with the simulation. In practice, how to use domain knowledge to pick the parameter should be elaborated, as it is an important factor.



**Q5 Detailed Comments To The Authors:**

1. The method is heavily dependent on the constraint set for the max player, which is constructed based on prior knowledge about the odds ratio.The constraint set, as mentioned by the authors, is very hard to specify as in DRO methods. The paper claims that their constraint set is an intuitive one. It certainly is, when we directly specify the ratio. But this is also very restrictive. It seems this framework can only work with a specific type of constraint set.

2. I do not quite understand why the parameter of policy is minimizing the (control variates of) the expected reward. Intuitively the parameter of the optimal policy should be maximizing the reward?

**Q7 Justification For Your Score:**

This paper proposes a novel method for off-policy policy learning. The method is novel and the writing is clear, but there are some concerns in the design and experiments.

**Q9 Complying With Reviewing Instructions:**

1: Yes.

---

### Decision · Program_Chairs · 2022-05-15

**Decision:**

Accept (Poster)

**Comment:**

Meta Review: In this paper, the authors consider the problem of learning decision rules (optimal with respect to some target distribution), where samples are drawn from a training distribution that potentially differs from the target distribution.  The difference between distributions is modeled using a selector variable S.  Since only the training data is observed (corresponding to the selector variable assuming the value 1), the authors treat the probability weight that would allow adjustment of policy learning to the target distribution to be a non-identified quantity.  Thus, rather than computing this weight directly, the authors choose to learn the worst case optimal policy (in the mini-max sense), using ideas from sensitivity analysis in causal inference where the sensitivity parameter bounds on the odds ratio quantifying the relationship of numerator and denominator of the selector weight.  Finally, the authors evaluate their methods on both synthetic and real data.

The reviewer consensus that emerged after the author feedback phase and discussion was positive overall, although some clarifications regarding proofs was requested for the final version of the paper.